# The bicentenary of Georg Hartung, a German pioneer geologist, explorer and illustrator

Carlos A. Góis-Marques[1,2], Miguel Menezes de Sequeira[1,4], José Madeira[2,3]

[1]Madeira Botanical Group (GBM), Faculdade de Ciências da Vida, Universidade da Madeira, Campus da Penteada, 9000-390 Funchal, Portugal
[2]Instituto Dom Luiz (IDL), Laboratório Associado, Universidade de Lisboa, Campo Grande, 1749-016 Lisboa, Portugal
[3]Departamento de Geologia, Faculdade de Ciências da Universidade de Lisboa, Campo Grande, 1749-016 Lisboa, Portugal
[4]CIBIO, Centro de Investigação em Biodiversidade e Recursos Genéticos, InBIO Laboratório Associado, Pólo dos Açores, 9501-801 Ponta Delgada, Portugal

*Correspondence to*: Carlos A. Góis-Marques (c.goismarques@gmail.com)

**Abstract.** We present a tribute to Georg Friedrich Karl Hartung (1821-1891), a less-known non-academic German geologist, on his 200[th] anniversary. Influenced by eminent 19[th] century scientific personalities such as Oswald Heer, Charles Lyell, and Alexander von Humboldt, he performed pioneer geological observations and sampling in the Azores, Madeira, and Canaries volcanic archipelagos. Later in his life he travelled to the USA and explored the Scandinavian countries. His scientific endeavours were published in several books and papers, many of them co-authored by academic German geologists and palaeontologists. His works on the macaronesian islands are deemed as classics, many enriched by his own detailed geological illustrations.

## 1 Travels, influences and published literature

Georg(e) Friedrich Karl Hartung or Georg Hartung, was born on the 13[th] of July of 1821 in Konigsberg (then Prussia, now Kaliningrad, Russia), and perished in Heidelberg (Germany) on the 28[th] of March 1891 (e.g. Bouheiry, 2015; Lindemann, 1891; Pinto and Bouheiry, 2007). He was born in a well-stablished and wealthy family, and his father owned a printing and a publishing company, being responsible for the publication of the journal 'Hartungsche Zeitung' (Bouheiry, 2015). To date no known portrait is available (Pinto and Bouheiry, 2007; A. Bouheiry, personal communication), although he represents himself in his geological drawings taking notes/sketching (Fig. 1A), or talking to local people (Fig. 1B) Hartung signature (Fig.2), was found within an unpublished manuscript (see Lyell, 1854/1855) in a partial and undated letter to Charles Lyell (1797-1875).

Most information about Hartung comes from his own work or correspondence (Pinto and Bouheiry, 2007; Bouheiry, 2013, 2015; Sarmiento Pérez, 2004). Hartung received a degree in Agronomy (1841-1843, University of Greifswald) (Bouheiry, 2015). Lacking a formal geological education, in 1855 he took private geology lessons with Gustav von Leonhard (1816-1878) (e.g. Pinto and Bouheiry, 2007; Bouheiry, 2013) and in 1862, he receives an honorary doctorate from the

University of Konigsberg, most probably due to his geological works on the Atlantic islands (Reifs, 1891; Bouheiry, 2015, 2013). Moreover, he was a gifted illustrator considering his geological sketches (see Figs. 3 to 8).

In the winter of 1850/51, suffering from catarrh, Hartung travelled to Madeira Island (Portugal) to seek a milder climate to ameliorate his poor health condition (Pinto and Bouheiry, 2007). In Funchal, he met the Swiss palaeobotanist Oswald Heer
(1809-1883), who travelled to Madeira due to similar health reasons (e.g. Schröter and Heer, 1885; Bouheiry, 2013). During this stay, Hartung accompanied Heer fieldwork (Heer, 1857; Schröter and Heer, 1885). Heer influenced him to start studying the geology of Madeira Island (Bouheiry, 2013). Hartung returned to Madeira Island in the following winter (1851/52), also traveling to Porto Santo (Madeira archipelago) and Tenerife (Canary archipelago) to make geological observations (Bouheiry, 2013). On his return to Madeira in the winter of 1853/54, he met the English geologist Charles Lyell (1797-1875) that had
travelled to Madeira to study the geology of the Atlantic Islands. According to Wilson (2007), Lyell went to Madeira to evaluate the catastrophist theory of 'craters of elevation' postulated by Leopold Von Buch (1774 – 1853) (Buch, 1826), against his uniformitarian views (Lyell, 1855). Lyell choose Hartung to accompany him due to his geological curiosity, knowledge of the Portuguese language, and the geography of the island. Both travelled to Madeira and to the Canary Islands of Tenerife, La Palma and Gran Canaria (Pinto and Bouheiry, 2007; Wilson, 2007). In the winter of 1854 Hartung proceeded alone to explore
the Canary Islands of Lanzarote and Furteventura (Bouheiry, 2015).

Hartung planned to publish the geological observations of Madeira and the Canaries archipelagos with Lyell, but due to Lyell's schedule this never came to fruition (Bouheiry, 2013), although a manuscript draft by Lyell with several sketches and illustrations by Hartung does exist (see Lyell and Hartung, 1856, unpublished manuscript). In 1856, during a meeting with scholars in Berlin including Lyell, Hartung was encouraged by Alexander von Humboldt (1769-1859) to travel and study the
50 geology of the Azores Archipelago (Bouheiry, 2015, 2013) and he, in fact, travelled to the Azores archipelago in 1857. During his stay in the Azores, he produced a manuscript (Hartung, 1857a) on the geology of Terceira Island (see Pinto, 2007). Hartung's exploration in the Azores is mentioned in Charles Darwin (1809-1882) letters to Joseph Dalton Hooker (1817-1911) and to Lyell (Darwin Correspondence Project: letters no. 2262, 2263, 5183, 5185), and geological data reported to Darwin were included in the 'Origin of Species' first edition (see Darwin, 1959, pp. 363).

Having explored the archipelagos of Madeira, Canary and the Azores, he proceeds in 1858 to 1861 to explore the volcanic areas of Germany to compare them with his own observations in the Atlantic islands (Bouheiry, 2015). He started to publish his results and illustrations on Lanzarote and Fuerteventura (Hartung and Arlett, 1858; Hartung, 1857b; Fig. 3), on the Azores (Hartung, 1860a, b; Figs. 4 and 5) and on the Gran Canaria (Hartung, 1862). Hartung further publishes his observations on Madeira and Porto Santo Islands (Hartung and Mayer, 1864; Fig. 6 and 7), and is co-author of a book about the geology of
Tenerife (Fritsch et al., 1867; Fig. 8) being the last known drawings published. However, in 1870 he stills draws as he sends drawings to Heer depicting the Niagara Falls, and the village of Matt in Switzerland (see Bouheiry, 2013).

Between 1870-71 period, corresponding to the war between France and Germany, he travelled to the United States of America where he met Louis Agassiz (1807-1873) at Cambridge, Massachusetts (Bouheiry, 2015). Reports of this travel were published in his home journal 'Hartungsche Zeitung' (Bouheiry, 2015). In 1873 he changes his research focus, travelling

to Sweden, leading him to further explore the Scandinavian countries (Bouheiry, 2013, 2015). The results of these explorations were published in three books (Hartung and Dulk, 1877; Hartung, 1877a, 1882) and on a report on plant fossils (Hartung, 1877b). Last known writings appear mainly in the journal 'Zeitschrift der Gesellschaft für Erdkunde zu Berlin', where he publishes a review on the formation of lakes and valleys (Hartung, 1878a), a summary in german of the 'ninth annual report of the U. S. Geological and Geographical Survey of the territories' by Hayden (1875) (Hartung, 1879a), a translation to german

of a paper about the geography of Norway by Kjerulf (1876) (Hartung, 1879b), a description and discussion about the formation of the Jutulhogget Canyon in Norway (Hartung, 1880a), two summaries in german of the tenth (Hayden, 1878) and eleventh (Hayden, 1979) 'annual report of the U. S. Geological and Geographical Survey of the territories' (Hartung, 1881a, b), a discussion about the landslide in Flims, Switzerland (Hartung, 1884), and finally a summary in german of the 'third annual report of the United States geological survey' by Powell (1883) (Hartung, 1885). Further contributions were published in other

journals such as a commentary of the book 'scientific results of the United States arctic expedition' by Bessels (1876) (see Hartung, 1880b), and a discussion about the book 'The Great Ice Age: And Its Relation to the Antiquity of Man' by Geikie (1874) dealing with the glaciations (Hartung, 1878b).

        Hartung's own geological observations and certainly the influence of Lyell and Heer ideas, led him to provide detailed field evidence against catastrophic theories, pointing out in favour of the gradual upbuild of the Atlantic islands, the formation

of valleys by erosion or the uplift movements (Pinto and Bouheiry, 2007). Furthermore, he was also a pioneer in what concerns to collecting geological and paleontological specimens (but also entomological and botanical specimens; see Bouheiry, 2013, 2015), that were delivered mostly to German and Swiss academics for study and description, which were later published in Hartung's own books or in separate papers (e.g. Heer, 1857; Hartung and Mayer, 1864). This legacy allowed, in recent decades, to localize the fossiliferous outcrops described by Hartung almost 200 years ago and to obtain new and important data on the

geology of these sites and to collect new fossil specimens (e.g. Madeira et al., 2007; Góis-Marques, 2013; Góis-Marques et al., 2014; Góis-Marques et al., 2018; Góis-Marques et al., 2019).

        Despite the perseverance to travel in hard conditions and his publication track, Hartung is still a less-known figure in Geology, and further efforts are needed to put together the life of this explorer. Important data and clues to his scientific views and relations to other fellow geologist and other scientists are certainly to be found in his unpublished correspondence, namely

to Heer (Bouheiry, 2013) and in his reports to the 'Hartungsche Zeitung'. As pointed out by Pinto and Bouheiry (2007), the achievements of this German geologist should not be forgotten and should be celebrated as classic works on the geology of the Atlantic Islands.

**Data availability**. No data sets were used in this article.

**Author contributions**. All authors contributed equally.

**Competing interests**. The authors declare that they have no conflict of interest.

**Acknowledgments**

We would like to dedicate this work to Annete Bouheiry, for her work on Georg Hartung and for her kindness in providing information and bibliography; to Gillian McCay (Cockburn Geological Museum, University of Edinburgh), for the access to Lyell manuscript; to the reviewers Adriano Pimentel (University of Azores) and Karl-Heinz Glassmeier (Technische Universität Braunschweig) and to the editor Kristian Schlegel for their review and suggestions that improved the initial manuscript.

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

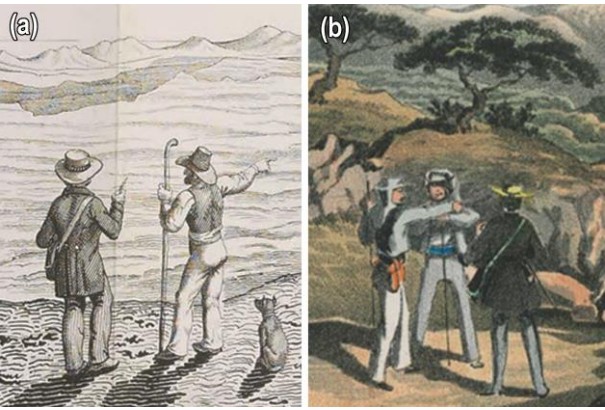

**Figure 1:** Details of two illustrations depicting Georg Hartung performing fieldwork. (a), G. Hartung (left figure with a hat) taking notes from his guide in Fuerteventura Island, Canary archipelago, extracted from plate III in Hartung (1857b; https://www.biodiversitylibrary.org/item/46730#page/320/mode/1up: last access 26 September 2021); (b), Same person in São Miguel (Azores archipelago) talking with the locals, extracted from plate VII in Hartung (1860a: https://www.e-rara.ch/zut/content/zoom/7111419: last access 26 September 2021). Both images are public domain.

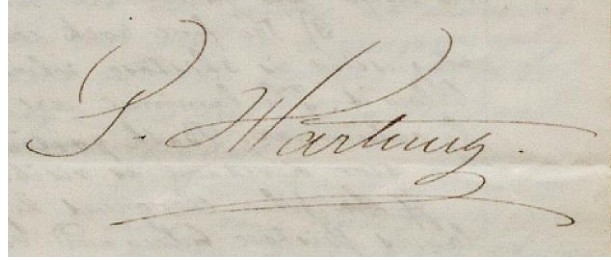

**Figure 2:** Signature of Georg Hartung found in a partial letter addressed to Lyell, within a Lyell's manuscript dealing with Madeira Island (see Lyell, 1854/1855). Image courtesy of the University of Edinburgh Library, under Creative Commons –attribution 4.0 International.

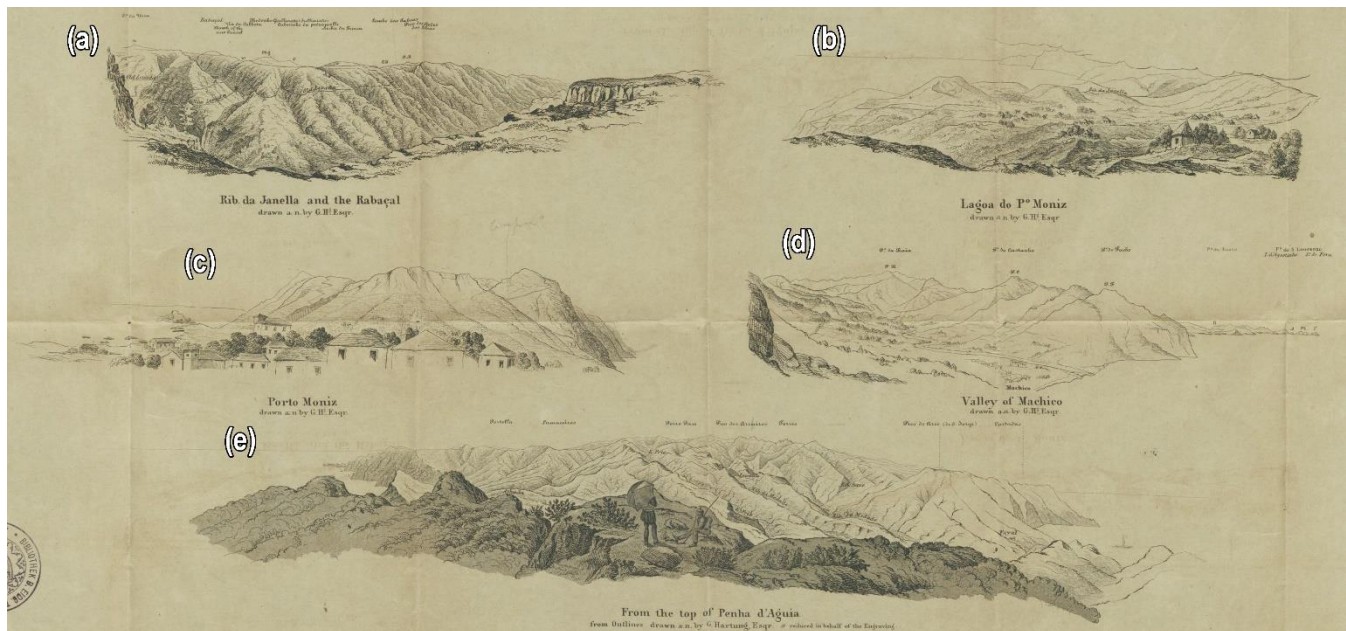

**Figure 3:** Panoramic views drawn by G. Hartung printed in the back side of Ziegler et al. (1856) physical map of Madeira Island. (a) Ribeira da janela stream and Rabaçal locality; (b) Lagoa locality in Porto Moniz; (c) Porto Moniz village; (d) Machico and Ponta de São Lourenço in the eastern part of the island; (e) View from the top Penha d'Águia (Porto da Cruz) showing the central part of the island. Notice that the depicted persons are probably Hartung wearing an umbrella while sketching, and his local guide waiting for him. Image public domain, digitalized by ETH-Bibliothek Zürich (https://doi.org/10.3931/e-rara-38937: last access 26 September 2021).

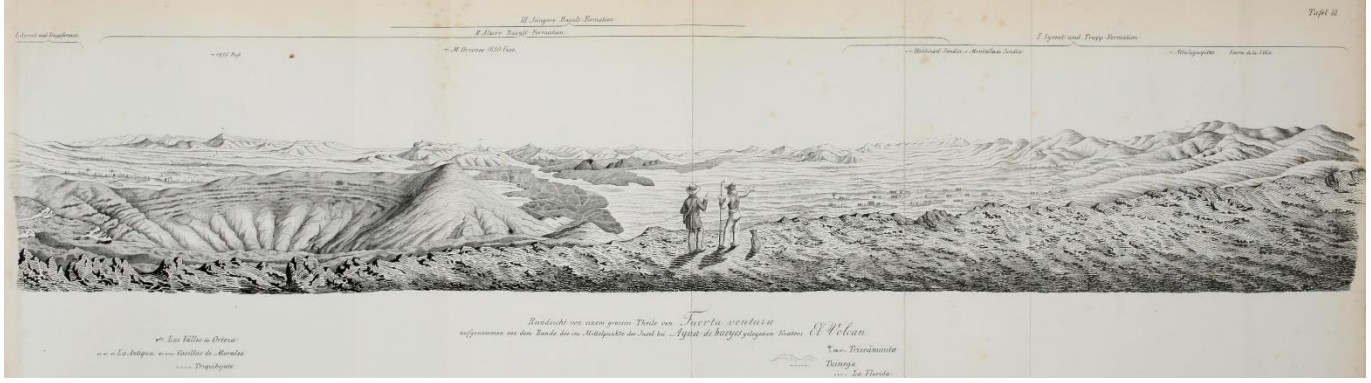

**Figure 4:** Hartung (1857b), plate III, depicting a panoramic view of Fuerteventura Island (Canary Archipelago) from the locality of Água de Bueyes. Image public domain, digitalized by Biodiversity Heritage Library (https://www.biodiversitylibrary.org/item/46730#page/320/mode/1up: last access 26 September 2021).

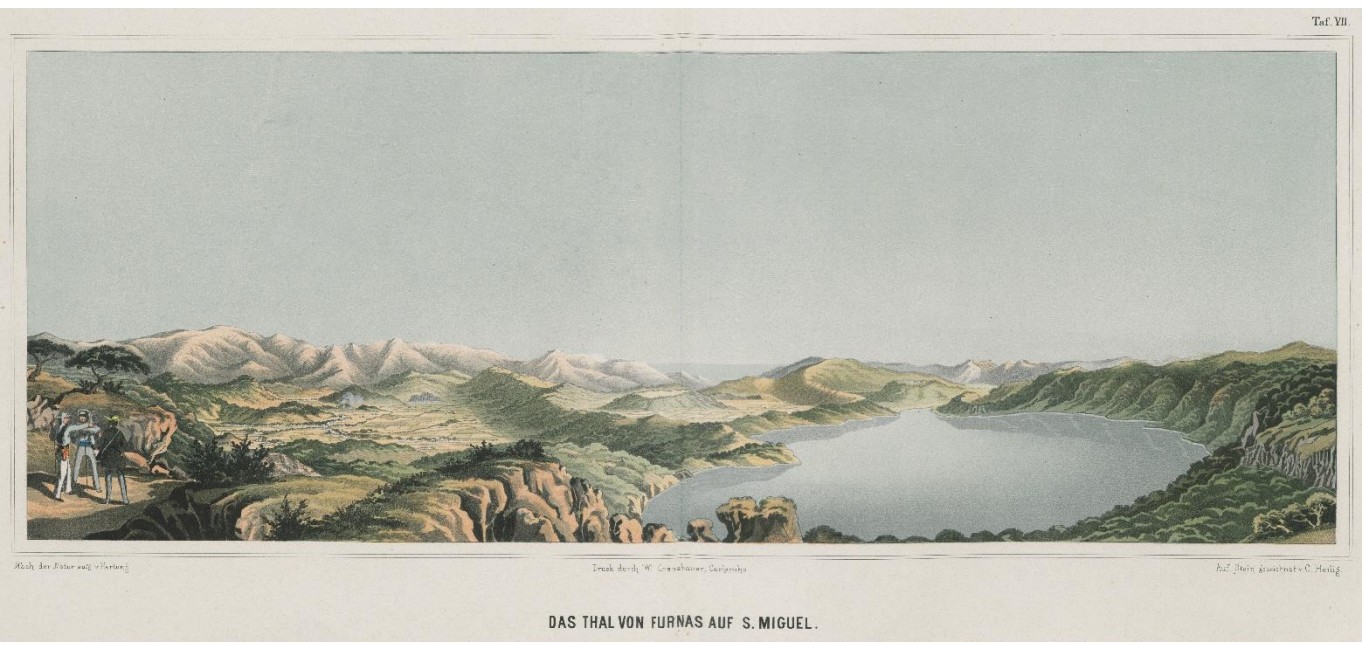

**Figure 5:** Hartung (1860a), plate VII, depicting a panoramic view of Furnas in São Miguel Island. Image public domain, digitalized by ETH-Bibliothek Zürich (available https://www.e-rara.ch/zut/content/zoom/7111419: last access 26 September 2021).

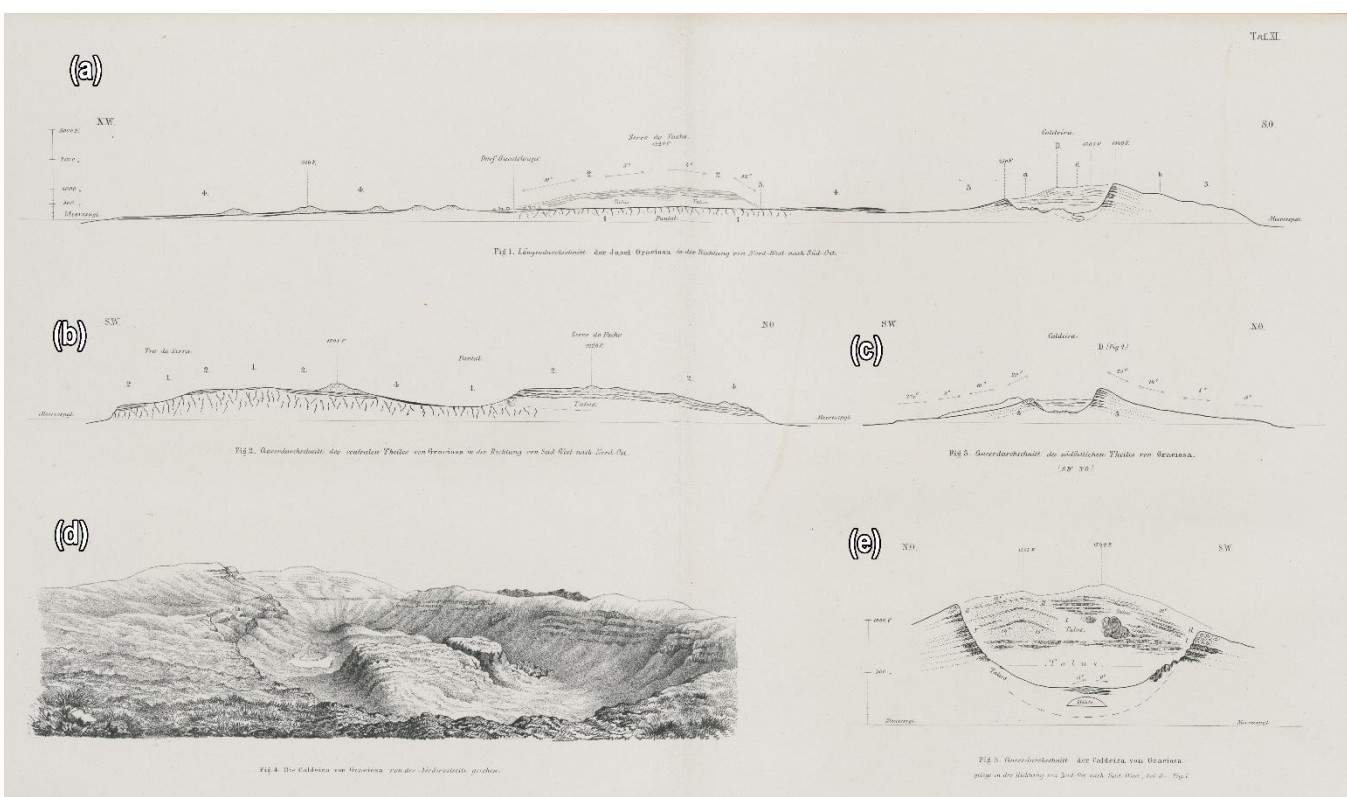

**Figure 6:** Hartung (1860a), plate XI depicting geological sketches of Graciosa Island, Azores archipelago: (a) NW-SE cross-section of the island depicting four main volcanic formations proposed by Hartung; (b) SW-NE cross-section of the island; (c) SW-NE cross-section of the southern part of the island, crossing the Caldera of Graciosa; d) panorama of Caldera of Graciosa; (e) NE-SW cross-section of Graciosa Caldera, signalling the Furnas do Enxofre ('hölhe' in the original). Image public domain, digitalized by ETH-Bibliothek Zürich (https://www.e-rara.ch/zut/content/zoom/7111423: last access 26 September 2021).

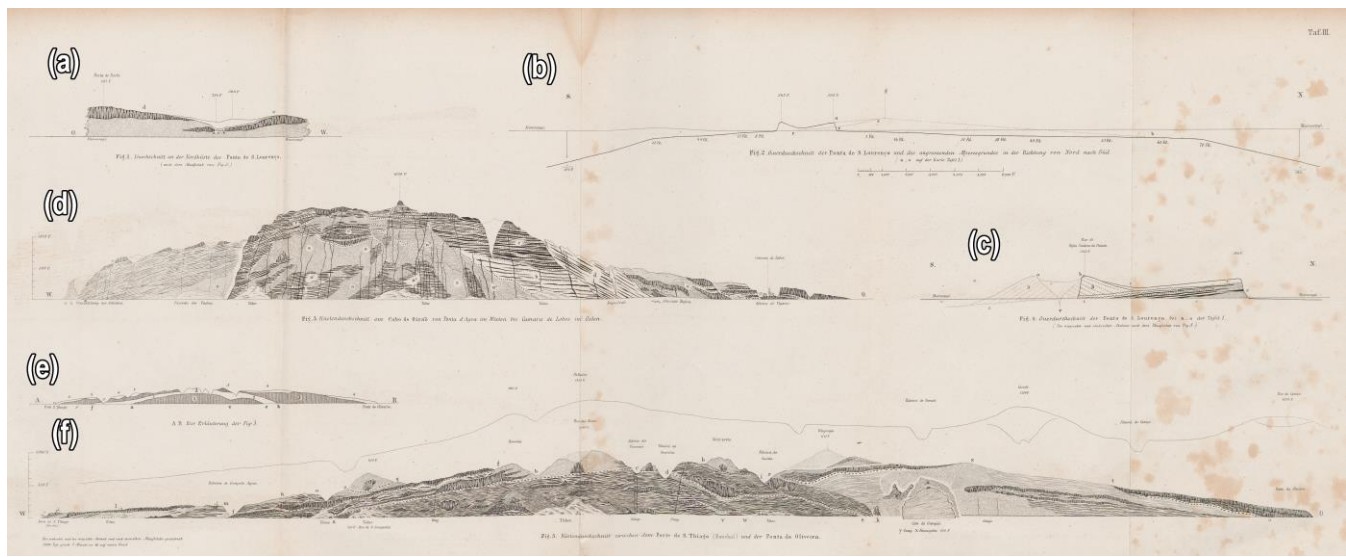

**Figure 7:** Hartung (1864), plate III, portraying geological sketches of Madeira Island, Madeira archipelago: (a), (b) and (c), sketches from the Eastern part of Madeira Island, the Ponta de São Lourenço: (a) section of the north cliff depicting the lithology; (b) N-S cross-section; (c) detail of the cross section in (b), including a stratigraphy and a reconstruction of the sea-eroded pyroclastic cone of Senhora da Piedade;

(d), (e) and (f), drawings of the south coast of Madeira Island: (d) sea cliff from Cabo Girão to Câmara de Lobos; (e) stratigraphy the sea cliff portrayed in (f); (f) sea cliff from Forte de São Tiago (Funchal) to Ponta da Oliveira (Caniço). Image public domain, digitalized by ETH-Bibliothek Zürich (https://www.e-rara.ch/zut/content/zoom/7222170: last access 26 September 2021).

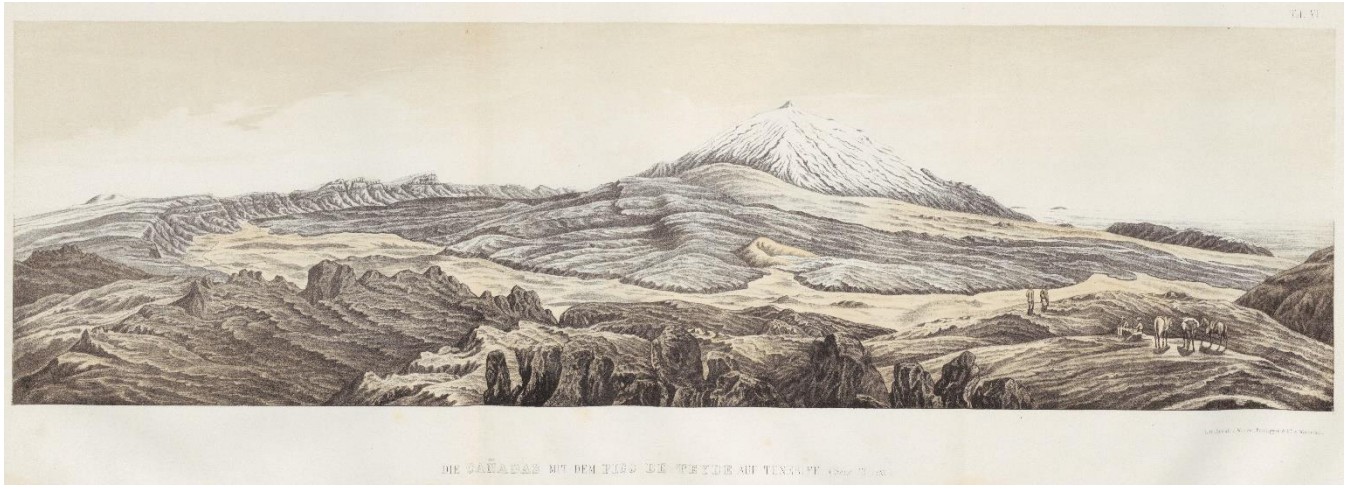

**Figure 8:** A Hartung drawing in Fritsch (1867), plate VI, portraying a view from Las Canadas into the Teide stratovolcano in Tenerife Island, Canary archipelago. A similar drawing was published in Lyell (1855, pg. 514, fig. 651), where credit is given to G. Hartung. Image public domain, digitalized by ETH-Bibliothek Zürich (https://www.e-rara.ch/zut/content/zoom/6096244: last access 26 September 2021).