# Peer review of "The bicentenary of Georg Hartung, a German pioneer geologist, and explorer and illustrator"

_History of Geo- and Space Sciences, 2021_

## Author Response (AR1)

**Response to Reviewers and Topical Editor**

**Reviewer 1**

I and my co-authors would like to thank Adriano Pimentel for the review. The suggestions and comments are indeed very relevant, and we changed our text accordingly to the proposed suggestions. In the following lines we present the reply to the comments and suggestions.

**RC1**. "I have only one major comment that could enrich this manuscript: Hartung's name is mentioned in the correspondence of Charles Darwin to Charles Lyell and Joseph Dalton Hooker. Given the paramount importance of Darwin to biological but also geological sciences, I think that adding a comment about this fact would highlight the importance of Hartung's work. It seems that the geological observations made by Hartung in the Azores sparked the thought of Darwin."

**AC:** We agree with the reviewer that the work of Hartung was indeed important to shape the thought of Darwin, Lyell, and Hooker. We included in the text the following phrase: "Hartung's exploration in the Azores is mentioned in Charles Darwin (1809-1882) letters to Joseph Dalton Hooker (1817-1911) and to Lyell (Darwin Correspondence Project: letters no. 2262, 2263, 5183, 5185), and geological data reported to Darwin were included in the 'Origin of Species' first edition (Darwin, 1959, pp. 363)."

References added:

Darwin, C.: On the origin of species by means of natural selection, or the preservation of favoured races in the struggle for life, John Murray, London, 1859.

Darwin Correspondence Project, Letter no. 2262: https://www.darwinproject.ac.uk/letter/DCP-LETT-2262.xml, last access 9 September 2021.

Darwin Correspondence Project, Letter no. 2263: https://www.darwinproject.ac.uk/letter/DCP-LETT-2263.xml, last access 9 September 2021.

Darwin Correspondence Project, Letter no. 5183: https://www.darwinproject.ac.uk/letter/DCP-LETT-5183.xml, last access 9 September 2021.

Darwin Correspondence Project, "Letter no. 5185," https://www.darwinproject.ac.uk/letter/DCP-LETT-5185.xml, last access: 9 September 2021.

**RC1.** "Besides that, there are only a few minor issues that need to be addressed before the manuscript can be considered for publication (see specific comments)."

Line 35: add "to" in "Madeira due to similar":

**AC**: We have followed the reviewer advice and added this in our revised manuscript.

Line 38: the authors should add that Porto Santo is an island of Madeira archipelago, as it is written one might think it is in the Canary Islands

**AC**: We changed the phrase to "also traveling to Porto Santo (Madeira archipelago) and Tenerife (Canary archipelago)"

Line 43: delete "(Spain)" it is already mentioned above

**AC:** We have followed the reviewer advice and deleted the word "Spain"

Lines 45-46: the sentence "Lacking a formal geological education, in 1855 he took private geology lessons with Gustav von Leonhard (1816-1878) (Bouheiry, 2013)." is a repetition of line 29. The authors could omit here as this paragraph mainly focuses on his travels and expand in line 29

**AC:** We agree with the reviewer. To avoid repetition, we placed this phrase in line 29 as suggested. Moreover, we deleted the repetition.

Lines 52 and 53: add "the" in "the Azores"

**AC:** We have followed the reviewer advice and added this in our revised manuscript.

Lines 57: replace "in" by "of" in "co-author of a book"

**AC**: We have followed the reviewer advice and added this in our revised manuscript.

Lines 57-28: the sentence "Due to his work, he receives in 1862 an honorary doctorate from the University of Konigsberg (Reifs, 1891)." is a repetition of line 30. The authors could omit here as this paragraph mainly focuses on his publication record and expand in line 30.

**AC:** We have followed the reviewer advice and deleted this from our revised manuscript

Line 65: it seems that something is missing to connect both parts of the sentence, before and after the citations

**AC:** We agree with the reviewer. We modified the phrase in our manuscript to:'Last known writings appear mainly in the journal 'Zeitschrift der Gesellschaft für Erdkunde zu Berlin' (Hartung, 1878a, 1879a, b, 1880a, 1881a, b, 1884, 1885), although further geological discussions were published in other journals (see Hartung, 1880b, 1878b).'
* * *
**Reviewer 2**

I and my co-authors would like to thank Karl-Heinz Glassmeier for the review of our manuscript. The reviewer commentary questions the novelty of our work and proposes the change of the manuscript title. In the following lines we present the reply to these comments and suggestions:

**RC2.** "I also fund a very extensive Wikipedia entry about Georg Hartung with very similar information as provided in the present article. Also, the contribution by Pinto and Bouheiry (2007) is very detailed already. Therefore, I did not learn much more when reading the present contribution as compared to the already available information in previous publications and the web."

**AC:** The writing of this short tribute is to celebrate the 200[th] anniversary of Georg Hartung, and call attention to his work, especially in the Macaronesian Islands. To achieve this, we not only brought all pertinent bibliography written in the last decades, but we also sought and present a

complete list of his contributions, which cannot be found in other published or online resources (e.g., Wikipedia). Moreover, we cite at least two coeval obituaries never cited in other papers dealing with Hartung. Pinto & Bouhery (2007), although a fantastic contribution, lacks new details about Hartung, for instance, who inspire him to go to the Azores (Von Humboldt), among others. These details were written in two books in German with limited circulation (Bouhery 2013; Bouhery 2015). Our paper intends to produce not only a tribute, but also to digest all English, Spanish and German (and some Portuguese and French) bibliography into an open access paper in English. Finally, with our manuscript we want to call attention that more work could be performed to understand the contributions of this german geologist, especially in the unpublished letters to Oswald Heer and other 19[th] century scientists.

**RC2:** "I recommend to change the title into "A Tribute to Georg Hartung on the Occasion of his 200th Birthsday". This would indicate the main aim of the contribution."

**AC:** We appreciate the suggestion made. However, we opted to maintain title unchanged, as the 'type of paper' is already a 'tribute'. Adding the word 'tribute' in the title is in our opinion redundant.
* * *
**Topical Editor**

As Topical Editor I have examined the manuscript and the references carefully. Apart from a WIKIPEDIA entry, there exist already 3 publications about Georg Hartung which contain even more information than in the present publication:

1. Bouheiry (2013)

2. Bouheiry (2015)

3. Pinto and Bouheiry (2007)

Unfortunately (1) and (2) are in German and (3) is published in a book not easily accessible.

This the first reason for my decision below.

The manuscript constitutes an appropriate contribution commemorating the bicentenary of Georg Hartung. This is the second reason for my decision below.

Hartung was apparently a very skilful illustrator, many excellent illustrations in his publications prove this. (1) and (2) above contain none of his illustrations, (3) just one. The present publication contains several examples of Hartung's illustrations. - This is the third reason for my decision below.

The authors should put some more emphasis on Hartung's illustrations. They should not lump the illustrations together into one singe figure (as number 1) but separate them (in groups) with appropriate figure captions and may perhaps include even some more.

In addition, on page 3, line 65 the authors should name the subjects in the list of Hartung's publications and not just the years of publication.

After this revision in which the authors should also take into account the remarks of the other referees (as already indicated in their reply), I will recommend the publication of this manuscript in HGSS.

I and my co-authors would like to thank the topical editor Kristian Schlegel for the review and suggestions to our tribute paper to Georg Hartung. We have read the provided recommendations and changed or added new text accordingly. In the following lines we present the reply to the comments and suggestions.

**EC1**: The authors should put some more emphasis on Hartung's illustrations. They should not lump the illustrations together into one singe figure (as number 1), but separate them (in groups) with appropriate figure captions, and may perhaps include even some more.

**AC:** We agree with the editor suggestions. We split our Figure 1 in separate figures and add some more, making a total of eight figures. Moreover, we provide a figure with the signature of Georg Hartung. The new order and figure labels are as follows:

Figure 1: Details of two illustrations depicting Georg Hartung performing fieldwork. (a), G. Hartung (left figure with a hat) taking notes from his guide in Fuerteventura Island, Canary archipelago, extracted from plate III in Hartung (1857b; https://www.biodiversitylibrary.org/item/46730#page/320/mode/1up: last access 26 September 2021); (b), Same person in São Miguel (Azores archipelago) talking with the locals, extracted from plate VII in Hartung (1860a: https://www.e-rara.ch/zut/content/zoom/7111419: last access 26 September 2021). Both images are public domain.

Figure 2: Signature of Georg Hartung found in a partial letter addressed to Lyell, within a Lyell's manuscript dealing with Madeira Island (see Lyell, 1854/1855). Image courtesy of the University of Edinburgh Library, under Creative Commons –attribution 4.0 International.

Figure 3: Panoramic views drawn by G. Hartung printed in the back side of Ziegler et al. (1856) physical map of Madeira Island. (a) Ribeira da janela stream and Rabaçal locality; (b) Lagoa locality in Porto Moniz; (c) Porto Moniz village; (d) Machico and Ponta de São Lourenço in the eastern part of the island; (e) View from the top Penha d'Águia (Porto da Cruz) showing the central part of the island. Notice that the depicted persons are probably Hartung wearing an umbrella while sketching, and his local guide waiting for him. Image public domain, digitalized by ETH-Bibliothek Zürich (https://doi.org/10.3931/e-rara-38937: last access 26 September 2021).

Figure 4: Hartung (1857b), plate III, depicting a panoramic view of Fuerteventura Island (Canary Archipelago) from the locality of Água de Bueyes. Image public domain, digitalized by Biodiversity Heritage Library (https://www.biodiversitylibrary.org/item/46730#page/320/mode/1up: last access 26 September 2021).

Figure 5: Hartung (1860a), plate VII, depicting a panoramic view of Furnas in São Miguel Island. Image public domain, digitalized by ETH-Bibliothek Zürich (available https://www.e-rara.ch/zut/content/zoom/7111419: last access 26 September 2021).

Figure 6: Hartung (1860a), plate XI depicting geological sketches of Graciosa Island, Azores archipelago: (a) NW-SE cross-section of the island depicting four main volcanic formations proposed by Hartung; (b) SW-NE cross-section of the island; (c) SW-NE cross-section of the southern part of the island, crossing the Caldera of Graciosa; d) panorama of Caldera of Graciosa; (e) NE-SW cross-section of Graciosa Caldera, signalling the Furnas do Enxofre ('hölhe' in the original). Image public domain, digitalized by ETH-Bibliothek Zürich (https://www.e-rara.ch/zut/content/zoom/7111423: last access 26 September 2021).

Figure 7: Hartung (1864), plate III, portraying geological sketches of Madeira Island, Madeira archipelago: (a), (b) and (c), sketches from the Eastern part of Madeira Island, the Ponta de São Lourenço: (a) section of the north cliff depicting the lithology; (b) N-S cross-section; (c) detail of the cross section in (b), including a stratigraphy and a reconstruction of the sea-eroded pyroclastic cone of Senhora da Piedade; (d), (e) and (f), drawings of the south coast of Madeira Island: (d) sea cliff from Cabo Girão to Câmara de Lobos; (e) stratigraphy the sea cliff portrayed in (f); (f) sea cliff from Forte de São Tiago (Funchal) to Ponta da Oliveira (Caniço). Image public domain, digitalized by ETH-Bibliothek Zürich (https://www.e-rara.ch/zut/content/zoom/7222170: last access 26 September 2021).

Figure 8: A Hartung drawing in Fritsch (1867), plate VI, portraying a view from Las Canadas into the Teide stratovolcano in Tenerife Island, Canary archipelago. A similar drawing was published in Lyell (1855, pg. 514, fig. 651), where credit is given to G. Hartung. Image public domain, digitalized by ETH-Bibliothek Zürich (https://www.e-rara.ch/zut/content/zoom/6096244: last access 26 September 2021).

Moreover, we changed the title from 'The bicentenary of Georg Hartung, a German pioneer geologist and explorer' to 'The bicentenary of Georg Hartung, a German pioneer geologist, explorer and illustrator' to focus the title on his illustrations.

Other small addictions pertaining the focus on his illustrations:

Line 50: we added the following line: 'The first known published illustrations by Hartung are found in Madeira Island map by Ziegler et al. (1856) depicting several panoramas of the island (see Fig. 3).'

Line 63-65: Hartung further publishes his observations on Madeira and Porto Santo Islands (Hartung and Mayer, 1864; Fig. 6), and is co-author of a book with geological drawings of Tenerife (Fritsch et al., 1867; Fig. 8), being the last known drawings publishes. However, in 1870 he stills draws as he sends drawings to Heer depicting the Niagara Falls, and the village of Matt in Switzerland (see Bouheiry, 2013).

**EC1:** In addition, on page 3, line 65 the authors should name the subjects in the list of Hartungs publications and not just the years of publication.

**AC:** We agree with the author. We changed our text to include the subjects that Hartung wrote about, and when appropriate we provide the bibliography of the papers Hartung was discussing. We added the following text:

'Last known writings appear mainly in the journal 'Zeitschrift der Gesellschaft für Erdkunde zu Berlin', where he publishes a review on the formation of lakes and valleys (Hartung, 1878a), a summary in german of the 'ninth annual report of the U. S. Geological and Geographical Survey of the territories' by Hayden (1875) (Hartung, 1879a), a translation to german of a paper about the geography of Norway by Kjerulf (1876) (Hartung, 1879b), a description and discussion about the formation of the Jutulhogget Canyon in Norway (Hartung, 1880a), two summaries in german of the tenth (Hayden, 1878) and eleventh (Hayden, 1979) 'annual report of the U. S. Geological and Geographical Survey of the territories' (Hartung, 1881a, b), a discussion about the landslide in Flims, Switzerland (Hartung, 1884), and finally a summary in german of the 'third annual report of the United States geological survey' by Powell (1883) (Hartung, 1885). Further contributions were published in other journals such as a commentary of the book 'scientific results of the United States arctic expedition' by Bessels (1876) (see Hartung, 1880b), and a discussion about the book 'The Great Ice Age: And Its Relation to the Antiquity of Man' by Geikie (1874) dealing with the glaciations (Hartung, 1878b).'

Added bibliography:

Bessels, E.: Scientific results of the United States Arctic Expedition. Vol. I. Physical Observations, Government Priting office, Washington, https://archive.org/details/cu31924029881095, 1876.

Geikie, J.: The Great Ice Age: And Its Relation to the Antiquity of Man, D. Appleton and Company, New York, 545 pp. 1874, https://books.google.pt/books?id=zxcBAAAAYAAJ

Hayden, F. V.: Ninth annual report of the United States Geological and Geographical Survey of the Territories: embracing Colorado and parts of adjacent territories; being a report of progress of the exploration for the year 1875, Govt. Print. Off., Washington, https://www.biodiversitylibrary.org/item/124508, 1877

Hayden, F. V.: Tenth Annual Report of the United States Geological and Geographical Survey of the Territories, embracing Colorado and parts of adjacent Territories, being a report of progress of the exploration for the year 1876, Govt. Print. Off., Washington D.C., https://doi.org/10.3133/70038934, 1878.

Hayden, F. V.: Eleventh Annual Report of United States Geological and Geographical Survey of the Territories embracing Idaho and Wyoming, being a report of progress of the exploration for the year 1877, Govt. Print. Off., Washington D.C., https://doi.org/10.3133/70038936, 1879.

Kjerulf, T.: Et Stykke Geografi i Norge, Forhandlinger i Videnskabs-selskabet i Christiania., 1-18, https://www.biodiversitylibrary.org/item/149466#page/105/mode/1up, 1876.

Powell, J. W.: Third Annual report of the United States Geological Survey to the Secretary of the Interior, 1881-1882, Report 3, 718, https://doi.org/10.3133/ar3, 1883.

**AC:** We also made the following small changes:

Line 89-90: added the following references: Góis-Marques, 2013; Góis-Marques et al. 2014:

Góis-Marques, C. A.: Paleobotânica da Ilha da Madeira: Inventário e Revisão da Macroflora Fóssil de São Jorge e Porto da Cruz, Dissertação de Mestrado Departamento de Geologia, Faculdade de Ciências da Universidade de Lisboa, Lisboa, 144 pp., 2013.

Góis-Marques, C. A., Menezes de Sequeira, M., and Madeira, J.: Palaeobotany of Madeira Island: historical perspective of the leaf-beds and collections of S. Jorge and Porto da Cruz, Silva Lusitana, nº Especial, 87-108, 2014.

Line 105: we added the following text: 'to Gillian McCay (Cockburn Geological Museum, University of Edinburgh), for the access to Lyell manuscript; to the reviewers Adriano Pimentel (University of Azores) and Karl-Heinz Glassmeier (Technische

Universität Braunschweig) and to the editor Kristian Schlegel for their review and suggestions that improved the initial manuscript.'